# The Role of Amplitude-Integrated Electroencephalography (aEEG) in Monitoring Infants with Neonatal Seizures and Predicting Their Neurodevelopmental Outcome

**DOI:** 10.3390/children10050833

**Published:** 2023-05-03

**Authors:** Florina Marinela Doandes, Aniko Maria Manea, Nicoleta Lungu, Timea Brandibur, Daniela Cioboata, Oana Cristina Costescu, Mihaela Zaharie, Marioara Boia

**Affiliations:** Department of Neonatology, “Victor Babes” University of Medicine and Pharmacy Timisoara, Eftimie Murgu Square 2, 300041 Timisoara, Romania; doandes.florina@umft.ro (F.M.D.); lungu.nicoleta@umft.ro (N.L.); timea.brandibur@umft.ro (T.B.); cioboata.daniela@umft.ro (D.C.); costescu.oana@umft.ro (O.C.C.); mihaela.zaharie@umft.ro (M.Z.); boia.marioara@umft.ro (M.B.)

**Keywords:** newborn, neonatal seizure, aEEG, neurological outcome

## Abstract

Newborn monitoring in neonatal intensive care units (NICU) is mandatory, but neurological and especially electroencephalographic (EEG) monitoring can be overlooked or delayed until the newborn is clinically stable. However, the neonatal period is associated with the highest risk of seizures in humans, and the clinical symptoms may often be discrete, but the evolution and long-term neurodevelopmental disorders in these patients may be important. In response to this issue, we conducted a study to evaluate newborns who experienced neonatal seizures (NS) in the NICU and monitored their long-term neurological development. We enrolled 73 term and preterm newborns who underwent EEG monitoring using amplitude-integrated electroencephalography (aEEG). We then followed their neurological development until around 18 months of age, with 59 patients remaining in the long-term study. A total of 22% of patients with NS developed epilepsy, 12% cerebral palsy, 19% severe neurodevelopmental disabilities, and 8.5% died within the first 18 months of life. Our findings indicate that aEEG background pattern is a strong predictor of unfavorable neurological outcomes, with an odds ratio of 20.4174 (*p* < 0.05). Additionally, higher Apgar scores were associated with better outcomes (*p* < 0.05), with the odds of unfavorable neurological outcomes decreasing by 0.7-fold for every point increase in Apgar score. Furthermore, we found a statistically significant association between preterm birth and unfavorable neurological outcomes (*p* = 0.0104). Our study highlights the importance of early EEG monitoring in the NICU and provides valuable insights into predictors of unfavorable neurological outcomes in newborns who experienced NS.

## 1. Introduction

Due to immaturity, circulatory instability, hypoxia, hypoglycaemia or seizures, newborns from neonatal intensive care units (NICU) have a higher risk of brain complications and neurological sequelae [1,2,3]. The number of newborns at risk being discharged from the NICU is constantly growing due to improvements in intensive care techniques and experience in the field. Very high survival rates currently exist among newborns with life-threatening conditions in the past, for example, perinatal asphyxia, extreme prematurity, and congenital heart or lung malformations [4].

The neonatal period is associated with the highest risk of seizures in humans [5] and they are a major challenge for clinicians due to discrete clinical signs, variable electro-clinical correlation, and poor response to medication [6]. The reported prevalence and incidence of neonatal seizures (NS) varies according to gestational age (GA) and birth weight (BW), being estimated up to 58/1000 live very low birth weight (VLBW) preterm newborns and 1 to 3.5/1000 live term newborns [7,8,9]. Hypoxic-ischemic encephalopathy (HIE) is the most common condition that causes seizures in term newborns [10] and Apgar scores can provide useful prognostic data before other evaluations are available [11].

While a number of physiological parameters such as electrocardiogram (ECG), heart rate, blood pressure, oxygen saturation, or body temperature have long been integrated into NICU monitoring systems, electroencephalography (EEG), which directly reflects the functional state of the brain, has been used less commonly [3]. There are a number of reasons for this. EEG interpretation requires extensive training. Additionally, the clinical condition of NICU patients does not usually allow one to monitor and maintain the impedance and position of multiple electrodes on the scalp for prolonged EEG recording (periods exceeding a few hours) [3]. However, EEG monitoring is absolutely mandatory to avoid misdiagnosed or over-diagnosed NS, especially in preterm newborns with a wide range of uncoordinated movements and in full-term newborns with various pathologies [10,12,13]. The high percentage of subclinical seizures with an abundance of only electrographic seizures among newborns adds to this challenge [10,14,15].

Given these limitations, amplitude-integrated EEG monitoring (aEEG) has gained ground in neonatology because it is readily available at the bedside, less invasive, easy to set up, and allows for extended bedside recordings [16]. The EEG signal for aEEG is recorded from one channel or two channels with two or four symmetrically positioned parietal or frontoparietal electrodes, derived in two frontoparietal recording channels, one channel from each hemisphere [3]. When single-channel aEEG recordings are obtained, the recommended electrode locations are parietal P3 and P4 because they overlie the apices of the cerebrovascular watershed zones [3] and have been shown to detect more seizures than frontal electrodes [17]. When two-channel aEEG is used, the centroparietal electrode pairs C3/P3 and C4/P4 are most commonly recommended [17]. The method is based on EEG filtering and compression, attenuating frequencies below 2 Hz and above 15 Hz, semilogarithmic amplitude compression, rectification, smoothing, and compression over time, which allows for the long-term assessment of changes and trends in electrocortical background activity following a relatively easy-to-recognize pattern [16]. The amplitude has a linear display between 0–10 µV and logarithmic between 10–100 µV, increasing the sensitivity of the method for variations of background activity at small amplitudes [16]. The impedance of the electrodes is constantly checked by the device, and the signal is displayed at a speed of 6 cm/h [3]. aEEG electrodes can be placed by trained personnel from NICUs, without necessarily involving a trained EEG technician or specialist. Other benefits include the ability to continuously monitor and detect seizures, especially on devices with detection software included. The background information may be useful in determining the degree of encephalopathy, the effect of anticonvulsants, and the prognosis [18]. An important disadvantage is that the device does not cover the entire brain, so some focal discharges may escape recording [18]. Most seizures lasting >30 s can be identified when recording 1-channel biparietal aEEG. Concomitant recording of 5-channel EEG has shown that some short seizures, some focal discharges, or continuous peak discharges can be ignored when reading aEEG [16]. Some studies show that using two-channel aEEG, combined with the raw EEG path on the same monitor, improved the accuracy of interpretation [19,20,21].

According to current guidelines [22], in situations when and where EEG is not readily available, aEEG may be used, although its limitations are well recognized. Conventional EEG monitoring is considered the gold standard, provides a reliable diagnosis of NS, and seizures discharged on aEEG provide the probability of diagnosis and require further EEG confirmation when the patient’s condition allows it [22,23]. aEEG is effective in detecting prolonged seizures and status epilepticus [19,24]. A continuous aEEG pattern with amplitudes between 10 and 25 µV, with smooth or cyclic variations or with a fully developed sleep-wake cycle (SWC), is usually a reassuring sign of uncompromised brain function [16]. Additionally, the background electroencephalographic pattern and the seizure burden are important predictors of the individual outcome of patients with NS [7,25]. The presence of sleep-wake cycle (SWC) in aEEG recording in the neonatal period was associated with a better prognosis [25,26].

Given these data on the value of aEEG monitoring in the neonatal period, the aim of the present study is to identify possible predictors of neurological outcomes in patients with NS, including aEEG background pattern and the presence of SWC.

## 2. Materials and Methods

### 2.1. Study Design and Ethics

This prospective study followed the evaluation of newborns with NS admitted to Neonatology and Preterm Department of the “Louis Turcanu” Children’s Emergency Clinical Hospital Timisoara over a period of 3 years. The patients were clinically and electroencephalographically evaluated during the neonatal period and their neurological outcome was subsequently followed around the age of 18 months.

“Louis Turcanu” Children’s Emergency Hospital is associated with the “Victor Babes” University of Medicine and Pharmacy Timisoara (UMFVBT). The study was conducted with the approval of the Research Ethics Committee of the University of Medicine and Pharmacy “Victor Babes” of Timisoara, CECS Opinion no. 41/2017 and the approval of Ethics Committee for Scientific Research of the “Louis Turcanu” Children’s Emergency Clinical Hospital Timisoara no. 13/6.03.2023. Additionally, a written informed consent for the conduct of the study was obtained from the parents of the children included in the study.

### 2.2. Inclusion Criteria and Study Variables

The criteria for inclusion in the study in the neonatal period were as follows: term newborns up to 28 days old and preterm newborns whose corrected GA has not exceeded 44 weeks at the onset of seizures, with clinical diagnosis or clinical suspicion of NS, regardless of their etiology; aEEG monitoring for at least 24 h, instituted prior to the time of starting antiepileptic drug (AED) treatment.

Exclusion criteria were as follows: age at onset of seizures higher than that mentioned in the inclusion criteria; patients in whom aEEG monitoring was not instituted prior to the time of starting AED treatment; aEEG monitoring shorter than 24 h.

Data collected included: patient gender, GA, BW, seizure type, Apgar score, need for mechanical ventilation, age at seizure onset, aEEG background pattern at the start of recording and at 24 h, and presence or absence of SWC on aEEG.

The study group was divided according to GA in the following manner: term newborns formed one group (GA ≥ 37 weeks) and preterm newborns formed another group (GA < 37 weeks). Seizure events were classified according to the Neonatal Seizures Task Force 2020 Classification: clonic, tonic, myoclonic, autonomic, epileptic spasms [22].

### 2.3. aEEG Data

aEEG background pattern was assessed by a neonatologist experienced in its interpretation. The background pattern from the beginning of aEEG recording and the pattern observed in evolution at 24 h of monitoring was noted. The aEEG background pattern was interpreted as follows [3,27]: (a) Normal/near-normal background activity: CNV—Continuous Normal Voltage (continuous activity with the minimum amplitude of 5–10 µV and maximum of 10–25 µV, and present SWC) or DNV—Discontinuous Normal Voltage (the minimum variable amplitude < 5 µV and the maximum > 10 µV); (b) Abnormal pattern: BS—“Burst Suppression” (discontinuous route with the minimum amplitude between 0–2 µV and discharges with amplitude > 25 µV), CLV—Continuous Low Voltage (≤5 µV) or FT—Inactive, Flat Trace (isoelectric path < 5 µV); (c) Seizures/epileptic activity: Sz—Seizures (sudden increase in lower amplitude accompanied by increase in higher amplitude), and SE—Status epilepticus (continuous unremitting seizures lasting > 30 min).

SWC presence was checked. SWC in the aEEG is characterized by smooth cyclic variations, mainly of the minimum amplitude. Periods with broader bandwidth represent more discontinuous activity during quiet sleep (tracé alternant in full-term infants), and the narrower parts of the trace correspond to more continuous background during wakefulness or active sleep. We considered: (a) No SWC: no sinusoidal variations of the aEEG background; (b) Developed SWC: clearly identifiable sinusoidal changes with cycle duration ≥ 20 min [3].

We used the Olympic Brainz Monitor CFM (Cerebral Function Monitor) for aEEG. Cross-cerebral-two-channel monitoring was used (5 electrodes: located at P3, P4, C3, C4, according to the 10/20 system and the reference electrode placed at the left shoulder), with a linear display of low-voltage activity (0–10 µV) and logarithmic display of activity between 10–100 µV. aEEG displayed as a time-compressed trend at 6 cm/h, combined with the raw EEG path and seizure detection software included.

### 2.4. Long-Term Outcome

Patients were followed up until discharge from the hospital and then the long-term outcome was assessed by analyzing the hospital’s ATLAS database, including subsequent admissions and visits to the pediatric and pediatric neurology departments. Diagnoses of epilepsy, cerebral palsy, severe psychomotor or acquisition delay, or death were followed. Patients who were not found in the follow-up database were excluded from the long-term outcome study. We considered: (a) Favorable/good outcome—patients for whom the above diagnoses were not mentioned at post-discharge assessments in the hospital; (b) Unfavorable/poor outcome—patients with epilepsy, cerebral palsy, severe psychomotor delay, or those who were deceased by around 12–18 months of age.

### 2.5. Statistical Analysis

Data were collected and analyzed using Microsoft Excel 2019 and SPSS version 26 (SPSS Inc., Chicago, IL, USA). To assess the significance of the differences between groups, Student’s *t*-test or analysis of variance (means, Gaussian populations), Mann–Whitney U test, or Kruskal–Wallis (medians, non-Gaussian populations) and χ^2^ (proportions) tests were used. Continuous variable distributions were tested for normality using the Shapiro–Wilk test, and for equality of variance using Levene’s test. The strength of association between two continuous variables from non-Gaussian populations was evaluated using Spearman’s correlation coefficient. Sample-size calculation was performed prior to the study, aiming to provide a statistical power of at least 80% and a confidence level of 95%. The predictive value of the background trace obtained by aEEG during the first 24 h of recording was plotted on an ROC curve with 95% CI. Logistic regression was performed to determine the predictive factors. In this study, *p* < 0.05 was considered the threshold for statistical significance.

## 3. Results

### 3.1. Population Characteristics

The initial study group consisted of 73 patients. Of the patients included in the study, 36 (49.31%) were term newborns, 37 (50.68%) preterm newborns, 49 (67.12%) were boys, and 24 (32.87%) girls. GA ranged from 22 to 42 weeks, and BW was between 450 g and 4500 g. In terms of delivery mode, 61% of term infants were born naturally, compared to only 30% of preterm newborns. Considering this, statistically significant differences were found between the two groups (*p* = 0.007). A total of 54% of preterm newborns required invasive or non-invasive respiratory support, compared to 30.6% of term newborns with NS—a difference without statistical significance (*p* = 0.127), as seen in Table 1. In total, 39.7% of cases had seizures in the first 72 h of life, and for 50.7%, seizures started after the first week of life. Regarding the clinical manifestation of seizures, the majority of infants in the study had autonomic, subtle seizures (35.6%), or clonic seizures (31.5%). In the group of preterm infants, the majority of cases (40.5%) had autonomic seizures (according to the modified ILAE newborn classification) [22], followed in incidence by clonic (27%) and myoclonic (16.2%) seizures. In the group of term newborns, the predominant clinical presentation was clonic seizures (36.1%), followed by autonomic (30.6%), and tonic-type (13.9%).

Regarding the etiology of NS, 49% were due to HIE, 23% had structural causes (cerebral infarction, hemorrhage or malformations, hydrocephalus), 11% were infectious, 6.8% metabolic, and 1.4% genetic.

### 3.2. aEEG Results

Given the high percentage of newborns with NS secondary to HIE, an association was made between aEEG pathway and Apgar score. The aEEG trace was analyzed at the beginning of the monitoring and in progression, and the background pattern was also noted at approximately 24 h of EEG recording (Table 2). Seizure and status epilepticus patterns (Figure 1) were found at a similar rate in newborns with Apgar scores between 0 and 5 compared to newborns with scores greater than 6 (66% and 60.9%, respectively). At the same time, we found normal background (CNV, DNV) among newborns with Apgar <5 (36%). The aEEG pattern at 24 h of monitoring showed that newborns with NS and an Apgar score of 0–5 had normal/near-normal course in reasonable proportion—46% had CNV and 28% DNV.

### 3.3. Prognostic Value of aEEG in Long-Term Neurological Outcome

Following the computerized search of medical records in the medical unit’s online database, of the 73 patients evaluated for NS in the neonatal period, only 59 remained in the study. Their data were further analyzed and associations and correlations were sought between their long-term neurological outcome and the data collected from the neonatal period.

We analyzed the long-term evolution of patients according to the electroencephalographic background pattern observed during aEEG monitoring, the presence/absence of SWC, Apgar score, GA, and gender. For the aEEG background pattern, we used the results obtained from the dynamic analysis of the background trace, approximately 24 h after the start of the EEG recording and after the initiation of AED treatment. Table 3 presents the results of a logistic regression analysis predicting the neurological outcome of NS at 12–18 months of age.

The odds ratio of aEEG is 20.4174, suggesting that children with abnormal aEEG results are 20.4 times more likely to have an unfavorable neurological outcome compared to children with normal aEEG results (with a *p* value < 0.05). A higher Apgar score is associated with better outcome (*p* < 0.05). The odds of unfavorable neurological outcome decrease by 0.7-fold for every point.

The predictive measures of the logistic regression model for predicting neurological outcome in children with NS reveal promising results. The model shows an accuracy of 86.44%. Moreover, the specificity of the model is 88.00%, highlighting its ability to correctly identify 88.00% of children with favorable neurological outcome. The sensitivity of the model is 85.29%, showcasing its potential to accurately identify 85.29% of children with unfavorable neurological outcome. Additionally, the AUC of the model is 0.8935, indicating its effectiveness in distinguishing between unfavorable and favorable neurological evolution cases (Figure 2).

### 3.4. Patient Outcome

We examined the relation between GA and the long-term outcome. Table 4 indicates that there is a statistically significant association between preterm birth and neurological outcome at 12–18 months of age. Infants born prematurely were more likely to reach an unfavorable outcome (*X*^2^ = 6.57, *p* = 0.0104).

Data collected in evolution showed that 22% of patients with NS developed epilepsy, 19% neurodevelopmental disabilities, 12% cerebral palsy, and 8.5% of them died within the first 12–18 months of life.

## 4. Discussion

The neonatal period is associated with the highest risk of seizures in humans [5]. This is an important aspect to consider for studies on the long-term development of these children, and there are also results on animal models showing increased susceptibility to subsequent unprovoked seizures (epilepsy) [5,28]. The diagnosis of neurological impairment should be made as early as possible through the use of screening tests or an effective clinical examination to identify individuals at risk of developing a certain condition [1,2]. Despite advances in cardiopulmonary care, newborns continue to suffer from critical neurological complications. Up to 25% of patients from the NICU can be diagnosed with encephalopathy, seizures, and other neurological disorders [3]. Due to the continuous development of diagnostic tools, such as cEEG, aEEG, video EEG monitoring or early neuroimaging, which complete the clinical observation, the diagnosis of NS can be established more accurately [1].

aEEG is an increasingly used method for the continuous monitoring of electrical brain activity in critically ill newborns in the NICU [18,29]. The method has advantages as well as disadvantages. It is performed at the bedside, uses a small number of electrodes [4,30,31,32], and has the ability to continuously monitor and detect seizures, especially on devices with seizure detection software. An important disadvantage is that the device does not cover the whole brain; therefore, some focal manifestations may be missed [33,34].

aEEG is a reliable device for the correct diagnosis of NS and for monitoring the evolution under treatment [34,35]. aEEG recordings are evaluated mainly for background pattern and seizure activity detection [18]. When the condition of the infants allows, they should be evaluated via cEEG or videoEEG to confirm seizures observed on aEEG [22].

We monitored aEEG, observed the background pattern of newborns with NS and compared patient pathways by degree of neurological impairment at birth, as measured by Apgar score at 1 min of life. The Apgar score is a clinical indicator that reflects the newborn’s physical condition at birth. Perinatal asphyxia and other risk factors, such as severe infections, prematurity, and maternal analgesia, may cause a low Apgar score and the need for cardiopulmonary resuscitation at birth [29,36]. Apgar scores provide useful prognostic data before other evaluations are available for infants with HIE [11]. At the beginning of the aEEG recordings, 40% of patients with NS and an Apgar score between 0 and 5 exhibited seizure patterns on a normal/near-normal background pattern, and 22% status epilepticus. In 36% of them, no seizure discharges were observed on aEEG trace, although they had clinical seizures. Their aEEG recording showed CNV or DNV trace. The widened trace of DNV indicates increased variability in background brain activity due to intermittent levels of lower activity [16,27]. In comparison, patients with NS and an Apgar score above 5 had electro-clinical seizures in 60.9% of cases, and 30.4% had normal trace (CNV, DNV). The normal or near-normal background pattern in patients with low Apgar scores can be explained by the delayed monitoring of patients who initially showed no clinical signs of seizures, knowing that the EEG background pattern of the newborns with HIE can turn to a normal/near-normal route in hours or days, depending on the degree of hypoxic-ischemic impairment [37,38].

Based on the fact that data found in the literature indicate that following these patients in dynamics is particularly important, as the transition to a normal background pattern is possible and with favorable long-term prognosis [37,38], we followed the progression of aEEG recordings in these patients. After about 24 h recording, we observed a normal background pattern in 74% of cases that had NS and an Apgar score of 0 to 5, and seizure discharges in a small percentage—2% of patients. A total of 22% of them presented EEG evolution towards a flat or microvolted pattern (FT or CLV), and 2% burst suppression (BS). CLV pattern is indicative of the brain spending the majority of its activity in the very low voltage range, and FT pattern could be indicative of the brain spending nearly all of its time with extremely low or no activity [16,27]. The BS pattern is indicative of the brain going through bursts of brain activity followed by periods of suppression which can be referred to as the interburst interval [16,27]. BS represents a severe abnormal pattern, which is encountered in brain malformations or severe asphyxia [22]. For patients with an Apgar score over 5, aEEG recording evolved to CNV and DNV trace in 60.9%, CLV 30.4%, and FT and BS 4.3%. Among the newborns with an Apgar score greater than 5, there were no cases with seizure discharge on aEEG at 24 h of recording. We did not observe significant differences between the aEEG background pattern of patients with an Apgar score higher or lower than 5 (*p* = 0.292, 0.580, respectively). However, the logistic regression analysis suggested that Apgar score is a significant predictor of unfavorable neurological outcome in children with NS. Higher Apgar scores were associated with better outcomes (*p* < 0.05), with the odds of unfavorable neurological outcomes decreasing by 0.7-fold for every point increase in Apgar score.

EEG monitoring is particularly important in the NICU as it provides important diagnostic and prognostic information [18,30]. Several studies have found that the background EEG pattern correlates well with the outcome of patients, both full-term and preterm newborns with seizures. Most newborns with seizures on a normal EEG background pattern have a good overall performance, while 90% of newborns with seizures on an abnormal EEG background pattern (low voltage, burst suppression) have an unfavorable outcome [6,39]. Moderate background disorders, which generally account for approximately 15% to 30% of traces, are associated with an intermediate probability of neurological sequelae [6]. Recovery of the background pattern to normal is not uncommon, and the sooner the background pattern normalizes, the better the prognosis [37,38]. We correlated the aEEG background trace dynamics 24 h after the start of the recording with the long-term neurological status of the patients remaining in the study. The logistic regression analysis suggests that an abnormal background pattern (BS, CLV, FT) is a strong predictor of unfavorable neurological outcome, with an odds ratio of 20.4174 (*p* < 0.05).

Neurological impairments such as ischemic, hemorrhagic, infectious, or paroxysmal discharges are frequently associated with subsequent motor, cognitive, behavioral, and sensory disorders [1,40]. In our study, evolving collected data indicated that 22% of infants with NS developed epilepsy, 19% severe neurodevelopmental disabilities, 12% cerebral palsy, and 8.5% of them died in the first 12–18 months of life. Data present in the literature show that early diagnosis can reduce mortality from 33% in 1990 to about 15% in 2005 and 10% in 2010 [41,42,43]. However, on the other hand, the prevalence of neuromotor sequelae remains relatively stable at around 27–55% [41,42].

Recently published studies show the advantage of using aEEG in preterm infants to predict long-term neurological evolution [44,45]. In our study, 78.3% of preterm infants with NS had an unfavorable outcome (cerebral palsy, epilepsy, sever psychomotor delay, or deceased) compared to 44.4% of term newborns. The differences were statistically significant (*p* = 0.0104).

The presence of sleep-wake cycle (SWC) in aEEG recording in the first week after birth was associated with a better prognosis, and the presence of seizures was not associated with a worse prognosis in these newborns [25,26]. Cyclic aEEG trace indicating SWC may also be present in some extreme preterm infants of 25 to 27 weeks gestation [25,26], and their presence in preterm infants under 32 weeks gestational age is helpful in predicting their survival [44]. In our study, the presence or absence of SWC on aEEG did not correlate with the long-term neurological outcome of patients (odds ratio is 0.3775, *p* > 0.05).

### Future Perspectives and Study Limitations

Further studies are needed to determine the neurological outcome of NS according to their etiology. The results of the study regarding the neurological outcome of patients with NS were based on the analysis of the hospital’s electronic program database and patients. Expert neurological assessment could not be properly and objectively performed during the COVID-19 pandemic, potentially impacting the results in terms of patients’ neurological outcome.

## 5. Conclusions

Our study highlights the importance of early EEG monitoring in the NICU and provides valuable insights into predictors of neurological outcomes in newborns with NS. Patients with an abnormal aEEG background pattern in dynamics (BS, CLV, FT) are more likely to have an unfavorable neurological outcome compared to newborns with normal aEEG results. A higher Apgar score is also associated with better neurological outcome in children with NS. In contrast, SWC did not prove to be a long-term prognostic factor in our study.

Preterm newborns with NS appear to have a higher risk of long-term adverse outcomes than term infants, and they have to be followed more closely for neurological impairment.

EEG monitoring should be started as early as possible in the NICU and requires dynamic assessment.

## Figures and Tables

**Figure 1 children-10-00833-f001:**
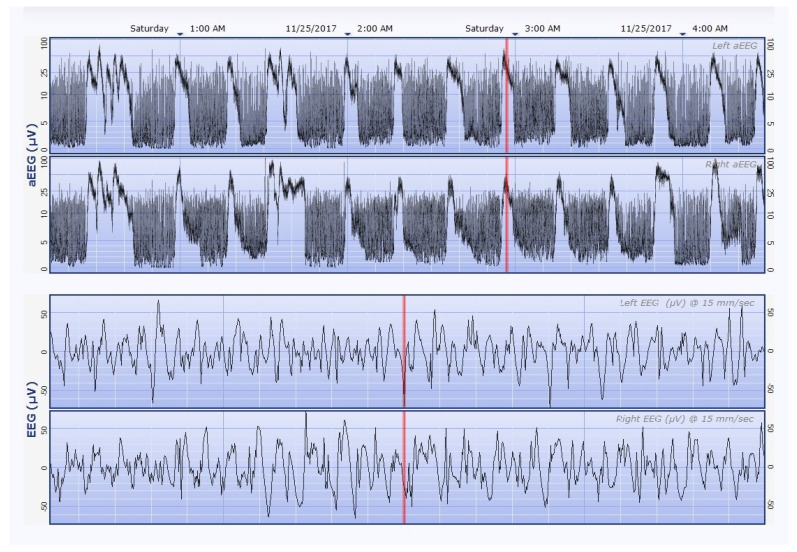
More than 3 h of aEEG (left, right aEEG) with 20 s of EEG (left, right EEG) from a preterm newborn with discontinuous background activity and many seizures ranging in duration between approx. 5 and 15 min. Seizures in aEEG trace are the sudden rise in the lower margin, accompanied by a rise in the upper margin. Additionally, we can observe rhythmic discharges using the EEG that is displayed below. The red vertical line shows the part of the aEEG from where the displayed EEG is taken. The two bottom panels show 20 s of EEG from the left and right side, respectively. The amplitude scales are shown on the left in µV. The thin narrow vertical lines through the aEEG denote 10 min intervals. In EEG recording, the thin narrow vertical lines denote 1 s intervals.

**Figure 2 children-10-00833-f002:**
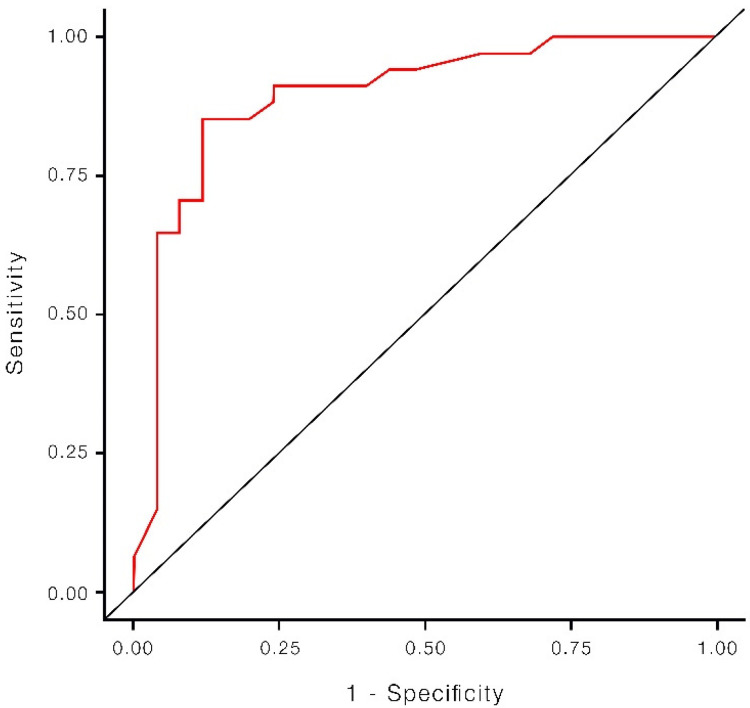
Predictive value of the aEEG background pattern, ROC curve.

**Table 1 children-10-00833-t001:** Characteristics of the group in terms of GA.

Newborn Features	Preterm (N = 37)	Term (N = 36)	Total (N = 73)	* p * Value
**Delivery**				0.007 *
Natural	11.0 (29.7%)	22.0 (61.1%)	33.0 (45.2%)	
C-section	26.0 (70.3%)	14.0 (38.9%)	40.0 (54.8%)	
** Mechanical ventilation **				0.127
Invasive ventilation	18.0 (48.6%)	10.0 (27.8%)	28.0 (38.4%)	
Non-invasive ventilation	2.0 (5.4%)	1.0 (2.8%)	3.0 (4.1%)	
** Age at onset of seizures **				0.234
0–3 days	13.0 (35.1%)	16.0 (44.4%)	29.0 (39.7%)	
3–7 days	2.0 (5.4%)	5.0 (13.9%)	7.0 (9.6%)	
>7 days	22.0 (59.5%)	15.0 (41.7%)	37.0 (50.7%)	
** Seizure type **				0.407
Clonic	10.0 (27.0%)	13.0 (36.1%)	23.0 (31.5%)	
Autonomic	15.0 (40.5%)	11.0 (30.6%)	26.0 (35.6%)	
Epileptic spasms	3.0 (8.1%)	5.0 (13.9%)	8.0 (11.0%)	
Myoclonic	6.0 (16.2%)	2.0 (5.6%)	8.0 (11.0%)	
Tonic	3.0 (8.1%)	5.0 (13.9%)	8.0 (11.0%)	

* *p* < 0.05.

**Table 2 children-10-00833-t002:** aEEG background pattern split by APGAR score.

Apgar Score	6–10 (N = 23)	0–5 (N = 50)	Total (N = 73)	* p * Value
** aEEG Trace Pattern **				0.292
CNV	4.0 (17.4%)	14.0 (28.0%)	18.0 (24.7%)	
DNV	3.0 (13.0%)	4.0 (8.0%)	7.0 (9.6%)	
CLV	2.0 (8.7%)	0.0 (0.0%)	2.0 (2.7%)	
Sz	10.0 (43.5%)	20.0 (40.0%)	30.0 (41.1%)	
SE	4.0 (17.4%)	11.0 (22.0%)	15.0 (20.5%)	
FT	0.0 (0.0%)	1.0 (2.0%)	1.0 (1.4%)	
** aEEG Trace Pattern Evolution at 24 h **				0.580
CNV	8.0 (34.8%)	23.0 (46.0%)	31.0 (42.5%)	
DNV	6.0 (26.1%)	14.0 (28.0%)	20.0 (27.4%)	
CLV	7.0 (30.4%)	7.0 (14.0%)	14.0 (19.2%)	
FT	1.0 (4.3%)	4.0 (8.0%)	5.0 (6.8%)	
BS	1.0 (4.3%)	1.0 (2.0%)	2.0 (2.7%)	
Sz	0.0 (0.0%)	1.0 (2.0%)	1.0 (1.4%)	

CNV—Continuous Normal Voltage, DNV—Discontinuous Normal Voltage, BS—Burst Suppression, CLV—Continuous Low Voltage, FT—Flat Trace, Sz—Seizures, SE—Status epilepticus.

**Table 3 children-10-00833-t003:** Predicting Neurological Outcomes at the age of 12–18 Months Using Logistic Regression: The Role of Gestational Age, aEEG Results, and Clinical Factors.

						95% Confidence Interval	
Predictor	Estimate	SE	Z	*p*	Odds Ratio	Lower	Upper
Intercept	2.4251	1.2205	1.987041	0.047	11.3034	1.03360	123.6137
EEG at 24 h:							
Abnormal–Normal trace	3.0164	1.2039	2.505411	0.012	20.4174	1.92840	216.1744
Seizure–Normal trace	18.0204	2399.5448	0.007510	NS	6.701 × 10^7^	-	-
Apgar score	−0.3228	0.1541	2.094674	0.036	0.7241	0.53534	0.9795
SWC							
Yes–No	−0.9742	0.7578	1.285469	NS	0.3775	0.08548	1.6672
Gender:							
F–M	−0.7834	0.8092	0.968171	NS	0.4568	0.09354	2.2312

Estimates represent the log odds of “Neurological evolution at 12–18 months = Unfavorable” vs. “Favorable”, NS—non-significant.

**Table 4 children-10-00833-t004:** Long-term outcome according to Gestational age (GA).

	Unfavorable Outcome (N = 34)	Favorable Outcome (N = 25)	Total (N = 59)	* p * Value
** GA **				0.0104
Preterm	18.0 (78.3%)	5.0 0 (21.7%)	23.0 (39.0%)	
Term	16.0 (44.4%)	20.0 (55.6%)	36.0 (61.0%)	

*p*-value was calculated by performing a χ^2^ test, GA—Gestational age.

## Data Availability

Data available on request.

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
