# Peer review of "The Role of Amplitude-Integrated Electroencephalography (aEEG) in Monitoring Infants with Neonatal Seizures and Predicting Their Neurodevelopmental Outcome"

_children, 2023, doi:10.3390/children10050833_

Round 1

Reviewer 1 Report

Thank you for letting me review this study. The idea of correlating aEEG with outcome is a good one, although similar work has been done in the past by Dr. De Vries' group and also by Dr. Renee Shelhaas.

The study will need major editing. The description of aEEG is inaccurate, the authors should mention all limitations of aEEG so the readers who will follow authors' advice for monitoring know the limits of the tests. One of the major disadvantages is lack of video which makes artifact identification harder. Also limited number of electrodes means seizures coming from other parts of the brain will be missed, so will be short seizures and low amplitude seizures. This should be mentioned.

The consensus in neonatal neuromonitoring and neonatal epilepsy world is that aEEG is a good screening test but full montage EEG should be done as soon as possible after suspected seizure in a newborn.

aEEG is a complementary too not the replacement for EEG. Authors should mention this, because currently it seems they are proposing aEEG as an easier alternative, that's incorrect.

As authors correctly pointed out, ICU/NICU is a hostile environment for aEEG/EEG monitoring because of sources of artifact - physiological and non-physiological. But authors are incorrectly implying that aEEG does make neuromonitoring easier which is incorrect. It is much harder to detect source of artifact on aEEG.

What are the etiologies of neonatal seizures in the newborns included in the study?

How quickly was aEEG monitoring started? In preterm and term neonates?

Every newborn was monitored for 24 hrs? when was it started? who made the decision.

Who made the decision to discontinue aEEG monitoring and what were the criteria to discontinue?

What do terms "unfavourable" or "favourable" mean - please specify what makes outcome favourable or not.

Sleep wake cycle has been shown to correlate with CP development by Dr. Shellhaas but authors can expand on this and add their preterm data to explore the field. Again, what measures and what outcomes were analyzed.

Reviewer 2 Report

In the present study, authors tried to evaluate the clinical value of aEEG monitoring of newborns with neonatal seizures and possibility of aEEG patterns to predict the outcome of these patients. It was a single center study on 73 patients.  Scientific and clinical relevance of the problem targeted in the study is high. However, the study presentation in the manuscript has major flows, from methodological and statistical perspectives to some style issues. Methodology is not clear in regard to study design and statistical analysis applied are not appropriate to draw desired conclusions. 

The study itself (collected data and results and idea) has a great potential, but it should be completely reorganized and rewritten.  

Additionally, the role of Notus who provided aEEG apparatus should be disclosed, as well as potential conflict of interest with this company (No external funding does not fit with donations) 

Some, among many, issues to be ameliorated are enlisted bellow. 

Major

Introduction

-      Introduction is not properly organized. It does not justify what will be the study goal. No explanation on aEEG. Authors said it is not commonly used in NICU, so how to expect that basis of aEEG is widely known without even one sentence of its explanations.  

-      The actual aim is not formulated properly. We could assume that the aim was to identify aEEG-based predictor factors of NS outcome in later infant age. Correlation is not adequate terminology here. 

Methodology 

-      Inclusion and exclusion criteria have to blearily stated

-      No data on calculation of aEEG parameters. Although, referred to other papers, a very brief description is still necessary in order to have smooth follow of the paper. 

-      “Patients who were not found in the database or for whom the above diagnoses were not mentioned were considered to have a favorable outcome.” This fact is not easy to accept. 

-      Statistical analysis descript is too general. It should be more specific. Are these methods appropriate to identify predictor factors,  i.e. predictive value of aEEG patterns? 

Minor points 

Abstract 

-First sentence in the abstract is too general and well known. Try to comibine it with the second one

-Abbreviation for SWC was introduced twice, but no introduction to aEEG related abbreviations. 

- Data on the sample size should be included in the abstract

- Report statistics of your findings. 

- Conclusion is too general and does not relay on the findings and does not answer the research question in the title (what about the role of aEEG in the conclusion?)

Introduction

-      Abbreviation introduction is not properly done. Eg. VLBW (very low body weight?), HIE, etc … 

Methodology

-      Written consent is not the same as informed consent. Without informed consent, this study is ethically inappropriate.

Conclusion

-      Some parts are too general and not supported by the study results. 

Round 2

Reviewer 1 Report

Thank you to the authors for significantly improving the manuscript.
I have a few minors comments:

1. can authors attempt to explain why newborns with lower Apgars had better aEEG tracings?

2. Why did authors use frontal and parietal electrodes? Standard is central and parietal since those cover watershed regions and areas where most seizures come from. Frontal seizures are rare in newborns. 

3. Could authors explain the abnormal neurodevelopmental outcomes with relation to aEEG results and Apgars? 

thank you 
